# Limited hyperoxia-induced proliferative retinopathy: A model of persistent retinal vascular dysfunction, preretinal fibrosis and hyaloidal vascular reprogramming for retinal rescue

Thomas Tedeschi[1], Kendal Lee[1], Wei Zhu[1,2], Amani A. Fawzi[1]*

1 Department of Ophthalmology, Feinberg School of Medicine, Northwestern University, Chicago, IL, United States of America, 2 Department of Ophthalmology, Changshu No. 2 People's Hospital, Changshu, China

* afawzimd@gmail.com

## Abstract

### Background

Retinopathy of prematurity (ROP) remains the leading cause for blindness in children. Limited hyperoxia induced proliferative retinopathy (L-HIPR) was recently introduced as a potential animal model for ROP and persistent fetal vasculature; however, the detailed pathological changes remain unclear.

### Methods

To model L-HIPR, we placed C57BL/6J mice in 65% oxygen from birth to post-natal day 7 (P7). We examined eyes at intervals between P12 and P30. Retinal morphometry, thickness, and preretinal fibrosis were quantified at different time points on histological sections stained with hematoxylin and eosin (H&E) and Masson Trichrome, respectively. Vascular development, angiogenesis, inflammation, and pericyte coverage were analyzed using immunohistochemistry staining in retinal flat mounts and cross sections.

### Results

In L-HIPR, the hyaloidal vessels persisted until the latest time point in this study, P30 and began to invaginate the peripheral then central retina starting at P12. Central retinal distortion was noted beginning at P17, while the peripheral retina demonstrated a trend of thinning from P12 to P30. We found that L-HIPR was associated with delayed and abnormal retinal vascular development with subsequent retinal inflammation, pericyte loss and preretinal fibrosis.

### Conclusion

Our study presents a detailed analysis of the L-HIPR animal model demonstrating vitreoretinal pathologic changes, preretinal fibrosis and persistent hyaloidal vessels into adulthood.

**Data Availability Statement:** All relevant data are within the paper and its Supporting Information files.

**Funding:** This work was supported by a grant from the National Institutes of Health (1R01EY030121, A.A.F.), and the Illinois Society for the Prevention of Blindness (KL). NO-The funders had no role in study design, data collection and analysis, decision to publish, or preparation of the manuscript.

**Competing interests:** The authors have declared that no competing interests exist.

Based on our findings, we suggest that the persistence and peculiar stepwise migration of the hyaloidal vessels into the retina may provide a potential rescue mechanism for inner retinal development that deserves further study.

## Introduction

To model ROP in rodents, the classic oxygen-induced retinopathy (OIR), originally introduced by Smith et al remains the most widely used and validated [1, 2]. In this model, neonatal mice are exposed to 75% oxygen for 5 days starting at postnatal day 7 (P7), whereupon they are transferred to room air at P12. During the hyperoxia exposure, the retinal vasculature undergoes vaso-obliteration, leaving large avascular retinal areas. Upon return to room air, the avascular neurons suffer relative hypoxia and ischemia, leading to up-regulation of hypoxia-inducible angiogenic factors with subsequent proliferation of pathological new vessels on the retinal surface. The pathologic neovascularization reaches its peak at P17 and then regresses gradually until it completely disappears by approximately P25, with relative normalization of the retinal vasculature [3]. While OIR provides a reliable and reproducible model for retinal neovascularization, it differs from clinical ROP in several aspects [4]. In human ROP, upon reaching the neovascular proliferative status, a subset of high risk eyes develop progressive pathology and advanced complications, which are visually detrimental unless surgical or medical treatment is implemented [5]. This is distinct from OIR, where as a rule, neovascularization regresses spontaneously. Retinal vessel attenuation and avascularity occur in the peripheral retina in ROP, while the avascular zone occurs in the central retina in the mouse OIR. Furthermore, OIR never progresses to severe complications, such as vitreous hemorrhages and retinal detachment, with regression and revascularization being the norm, unlike the course of clinically relevant, high risk ROP. Thus, there is a need for more relevant hyperoxia induced proliferative models to aid in our understanding of complicated ROP and other ischemic retinopathies.

McMenamin et al. [6] introduced a new, more chronic model of neonatal hyperoxia induced vitreoretinopathy, the limited hyperoxia induced proliferative retinopathy (L-HIPR), which showed prominent persistence of the hyaloidal vessels. These authors found that early exposure (P0-P7) to limited hyperoxia (65%) was associated with preretinal fibrotic scar formation, tortuous retinal vessels, hyperplastic and persistent hyaloidal vessels which remained visible up to 40 weeks postnatally. We hypothesized that these features could represent a relevant model for chronic ischemic retinopathy, similar to severe clinical ROP and advanced stages of diabetic retinopathy, where fibrovascular preretinal proliferation is prominent. The details of retinal and proliferative vitreoretinal pathology, including retinal layer disruption, retinal vascular changes as well as the potential role of hyaloidal vessels in this new model have not been fully explored.

We therefore sought to explore the retinal phenotype of the L-HIPR model in greater detail, using a combination of retinal morphometry and immunopathological studies. We found that in this model, the hyaloidal vessels provided an unexpected vascular rescue mechanism for the inner retina. This concept provides interesting and exciting potential avenues for future research aimed at coaxing the hyaloidal vessels to augment retinal vascular repair in ROP and other neonatal retinopathies.

## Materials and methods

### Animals

The experimental protocols in this study were approved by the Institutional Animal Care and Use Committee at Northwestern University and were conducted in accordance with the *Guide*

*for the Care and Use of Laboratory Animals of the National Institutes of Health.* A mixture of male and female C57Bl/6J (The Jackson Laboratory, Bar Harbor, ME) were used for all animal experiments in this study. Mice were genotyped to ensure the absence of the Crb1$^{rd8}$ mutation [7].

### Limited hyperoxia-induced proliferative retinopathy (L-HIPR)

The L-HIPR animal model was performed according to the protocol previously described by McMenamin et al. [6] Dams with their newly born litters (within 12 hours of birth) were placed in a Plexiglas chamber with an oxygen controller (ProOx 110; Biospherix, Lacona, NY, USA) and exposed to 65% oxygen from birth until postnatal day 7 (P7). Age-matched control pups remained at room air during P0-P7. Dams were rotated between hyperoxia and room air pups every 24 hours to prevent oxygen toxicity in these dams and loss of body weight in pups. With this approach, we achieved 100% survival of the dams, which also improved the pup survival to 70%. After 7 days of hyperoxia exposure, the pups and dams were returned to conventional cages and recovered at room air until experimental end points were reached.

### Histological preparation

For paraffin preparation, enucleated globes were fixed in 4% paraformaldehyde (PFA) overnight at 4˚C. After fixation, eyes were embedded in paraffin by the Mouse Histology and Phenotyping Core Laboratory of Northwestern University. Samples were then sectioned (7 μm) and stored at room temperature for further analysis. For cryosections, freshly enucleated globes were placed in Optimal Cutting Temperature (OCT) medium (Tissue-Tek, Torrance, CA) and flash frozen and stored at -80˚C. Samples were then sectioned (10 μm) and stored at -80˚C.

### Retinal thickness analysis

Samples were deparaffinized, rehydrated, hematoxylin and eosin (H&E) stained, and examined. For quantification, images were taken with Nikon 80i Eclipse microscope (Nikon, Tokyo, Japan) using a Photometrics CoolSnap CF camera (Photometrics, Tucson, AZ). An independent, masked investigator measured the thickness of each layer of the retina in three cross-sections/eye. The measurements were done using Image J software (National Institutes of Health, Bethesda, MD, USA) and the distance_between_polylines.java plug-in, as previously reported [8]. The average value for each layer (measured in 3 cross-sections/eye) was used in the final data analyses.

### Immunohistochemistry

Paraffin sections underwent antigen retrieval in sodium citrate buffer (10 nM sodium citrate, 0.05% Tween-20, pH 6.0) at 90˚C for 20 minutes. Cryo-sections were fixed with 4% PFA diluted in 1x Phosphate Buffered Saline (PBS; Invitrogen, Carlsbad, CA). Both types of sections were then blocked in 10% donkey serum (NDS)/0.1% Triton X-100/1x PBS for one hour at room temperature, incubated with primary antibody for 18 hours at 4˚C, and incubated with secondary antibody for one hour at RT. In order to quench photoreceptor autofluorescence, sections were stained 0.5% Sudan black diluted in 70% ethanol and counterstained with 4', 6-diamidino-2-phenylindole (DAPI; ThermoFisher Scientific, Carlsbad, CA) before being mounted with ProLong Gold Antifade reagent (Thermo Fisher Scientific, Waltham, MA). Sections were imaged with a Nikon A1R confocal laser microscope at the Northwestern University Center for Advanced Microscopy. See Table 1 for primary and secondary antibodies.

**Table 1. List of antibodies and stains used for immunohistochemistry.**

| Target | Source | Company | Product Number | Dilution | Tissue |
|---|---|---|---|---|---|
| α-smooth muscle actin (αSMA) | Rabbit | AbCam | ab5694 | 1:200 | FM |
| CD31 | Rat | BD Biosciences | 550274 | 1:50 | FM |
| Esm-1 | Goat | R&D Systems | AF1999 | 1:100 | FM |
| F4/80 | Rat | AbCam | ab16911 | 1:200 | IHC-Fr |
| GFAP | Rabbit | AbCam | ab7260 | 1:200 | IHC-Fr |
| Isolectin GS-IB4, Alexa Flour 568 Conjugate | | ThermoFisher | I21412 | 1:100 | FM, IHC-P, IHC-Fr |
| Isolectin GS-IB4, Alexa Flour 488 Conjugate | | ThermoFisher | I21411 | 1:50 | Cardiac Perfusion |
| NG2 | Rabbit | Sigma Aldrich | ZRB5320 | 1:500 | FM |
| Rhodopsin | Mouse | AbCam | ab5417 | 1:50 | IHC-P |
| Secondary Antibodies | Source | Company | Product Number | Dilution | Conjugate |
| Rabbit | Donkey | Jackson Immuno. | 711-295-152 | 1:200 | Rhodamine Red-X |
| Rabbit | Donkey | Jackson Immuno. | 711-605-152 | 1:200 | Alexa-647 |
| Rat | Donkey | Jackson Immuno. | 712-545-150 | 1:200 | Alexa-488 |
| Goat | Donkey | Jackson Immuno. | 705-545-147 | 1:200 | Alexa-488 |
| Mouse | Donkey | Jackson Immuno. | 715-545-150 | 1:200 | Alexa-488 |

Abbreviations: FM: Flatmount; IHC-P: Paraffin sections; IHC-Fr: Cryosections

### Retinal flat-mount analysis

Eyes were fixed in 4% PFA/PBS at room temperature for 2 hours. After dissection, the retinas were blocked with 5% normal donkey serum/1% bovine serum albumin (BSA)/0.3% Triton X-100/1x PBS for 1 hour at room temperature, labeled with **primary** antibody for 18 hours at 4°C, and incubated with secondary antibody for one hour. Samples were flattened on a slide by performing 4 radial incisions and mounted with ProLong Gold Antifade reagent (Thermo Fisher Scientific, Waltham, MA). Samples were imaged with Nikon W1 Dual CAM spinning disc confocal laser microscope at the Northwestern University Center for Advanced Microscopy. Retinal flat-mounts were prepared and volumetrically imaged with spinning disc microscopy. The blood vessels were classified into either the superficial, intermediate, or deep plexus and quantified along a 300μm optical line scans spanning the entire depth of the retina. To get an accurate measure of the vascular plexuses along their entire retinal length, line scans were taken near the optic nerve, the middle of the retina, and the retinal periphery. See Table 1 for primary and secondary antibodies.

### Cardiac perfusion assay

Mice were anesthetized deeply with a cocktail of ketamine: xylazine (100:20 mg/kg) via intraperitoneal (IP) injection. Once toe pinch reflex was absent, a 5-6cm lateral incision was made through the abdominal wall and the rib cage was cut to expose the pleural cavity. A 60μL 1mg/mL Alexa 488 isolectin GS-IB$_4$ conjugate (ThermoFisher, I21411) diluted in 1x PBS was injected directly into the left ventricle of the heart to label perfused retinal vessels. After five minutes the animal was euthanized, retinas were dissected and fixed for flatmount analysis using Alexa 568 isolectin GS-IB$_4$ conjugate (ThermoFisher Scientific, I21412) to label all retinal blood vessels.

### Quantifying vascular density and pericyte counts

P21 and P30 flat-mounts were labeled with NG2 (pericytes) and CD31(endothelial cells) (Table 1). Vessel length was quantified automatically in ImageJ (National Institutes of Health,

Bethesda, MD, USA) by thresholding and skeletonizing CD31-labeled vessels in the region of interest (ROI). NG2$^+$ pericytes were then manually counted by two masked, independent investigators (KL and TT). These measurements were done in 5 ROIs per eye. The graders examined a total of 60 ROIs in 12 eyes (3 L-HIPR or control eyes/timepoint). The vessel length and ratio of vessel length: pericyte count in L-HIPR were then normalized against the measurements in age-matched room air controls.

## Masson trichrome staining

Samples were deparaffinized, rehydrated, and stained with a Trichrome kit following manufacturer's instructions (ab150686, AbCam, Cambridge, UK). Images were taken with a Zeiss Axioskop/Nuance Camera at the Northwestern University Center for Advanced Microscopy.

## Statistical analysis

All data are presented as mean ± standard error of the mean (S.E.M.). Statistics was analyzed with SPSS software (v.27; IBM Corp, Armonk, NY). Biological triplicates from different litters were analyzed. The difference between continuous variables was analyzed by non-paired grouped t-test. Comparisons between three or more groups were performed using one-way ANOVA test with either Bonferroni or Tukey's post-hoc analysis. $p < 0.05$ was considered statistically significant.

# Results

## Long-term weight gain was suppressed in L-HIPR

We wanted to ensure there were no systemic consequences of L-HIPR treatment that could potentially confound retinal analysis. We therefore evaluated postnatal weight gain in L-HIPR neonatal mice. As shown in S1 Table, average litter weights steadily increased from P12 to P21 without significant difference between the room air controls and the L-HIPR group. After weaning at P21, the average weights continued to increase in both groups, although L-HIPR mice had a significantly lower weight at P30 compared to age matched room air controls ($p < 0.05$). Throughout these time points, the L-HIPR animals did not display any differences in developmental progression or behavior, as judged by ear flaps starting to come away from the head (~P3), milk spot disappearance (~P6), progression of fur coverage (~P7-10), and eyelid opening (~P13). We also noticed normal developmental behaviors of L-HIPR litters such as surface righting (~P4-5), starting to eat solid food on the floor of the cage (~P14), and venturing away from the nest (~P14).

## Retinal pathological changes and persistent hyaloidal vessels in L-HIPR

In cross-sections, we found that the hyaloidal vessels remained prominent until P30 in L-HIPR, compared to the room air group where they were no longer visible by P17 (Fig 1B). In addition, we found that hyaloidal vessels invaded the central and peripheral retina at P12 and P17, respectively. Sporadic vitreous hemorrhaging was observed starting at P17 through P30 in the L-HIPR animals. Retinal folding, rosettes and peripheral retinal detachments were also extensive at P30 in L-HIPR (Fig 1B). Interestingly, despite all these pathological changes, the retinal stratification appeared to be relatively well preserved in L-HIPR.

## Inner retinal thinning is observed in L-HIPR

To confirm our qualitative impression of retinal layer preservation, we performed morphometric quantification of central and peripheral retina thickness, as displayed in Fig 1C. In the central retina of L-HIPR between P12 and P30, we found that the hyperplastic hyaloidal vessels

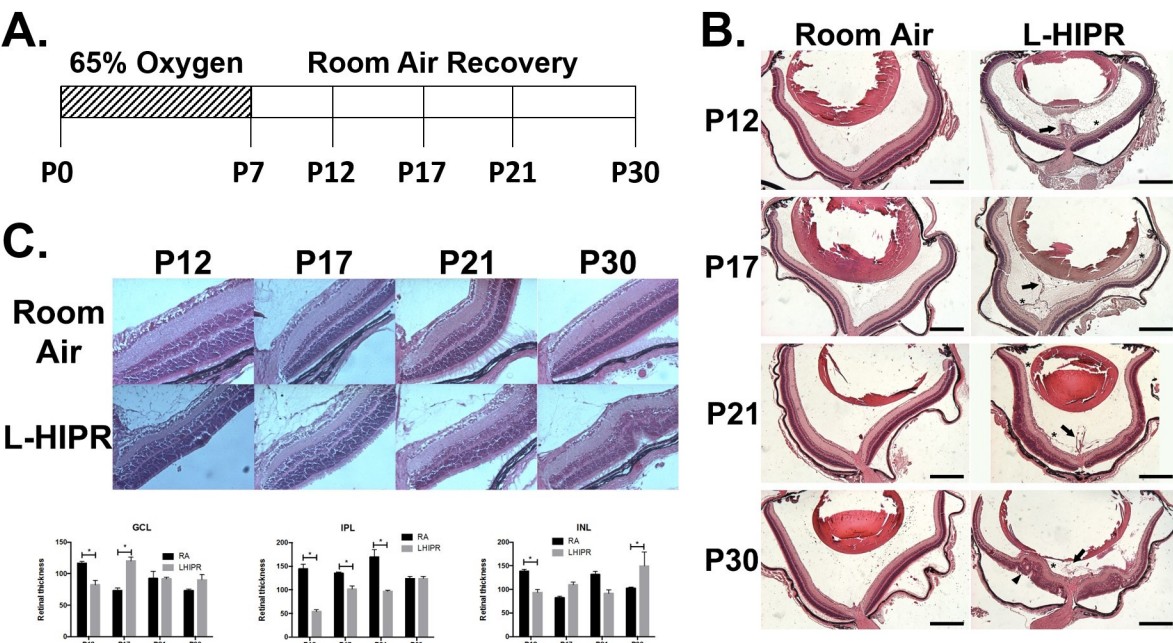

**Fig 1. L-HIPR induced retinal pathological changes and persistent hyaloidal vessels.** (A) Schematic of L-HIPR treatment of 65% oxygen from P0-P7 followed by room air recovery until experimental endpoints. (B) Paraffin embedded histological cross sections of P12, P17, P21 and P30 murine eyes in room air L-HIPR showing persistent hyaloidal vessels (arrow) since P12. Hyaloidal vessels associate with the central retina at P12 (asterisk) as well as the peripheral retina at P17 and P21. Retinal pathological changes started to appear at P30 including rosettes (arrowhead), detachments and intravitreal hemorrhaging. (C) Representative H&E-stained cross sections of central retina in room air and L-HIPR groups P12, P17, P21 and P30 (n = 3–4) where retinal layer thickness was quantified. Graphical representation of select layers of interest shown of the ganglion cell layer (GCL), inner plexiform layer (IPL), and inner nuclear layer (INL). * $p < 0.05$, one-way ANOVA with Tukey's post hoc analysis. Scale bars equal 500µm.

attached to the retinal ganglion cell layer (GCL) and elevated the inner retina surface. Compared to the room air group, the inner nuclear layer (INL), outer plexiform layer (OPL) and outer nuclear layer (ONL) were slightly distorted but were still identified from P17 through P30. Different from the somewhat disorganized central retina, the histological abnormalities were less prominent in the peripheral retina.

To further quantify the effect of L-HIPR on retinal development, we measured the thickness of GCL, inner plexiform layer (IPL), INL, OPL, ONL and photoreceptor (PR) in the central and peripheral retina at P12, 17, 21 and 30. As shown in Fig 1 and S2 Table, we found that the central GCL thickness in L-HIPR was significantly decreased at P12, while it was thicker at P17 relative to the room air group. There was also significant thinning of the central IPL at P12, P17 and P21 in L-HIPR. Thinning of the central INL was observed at P12 in L-HIPR. In the peripheral retina, the IPL was thinner at every time point except P17 in L-HIPR. Except for the peripheral ONL at P30, there were no significant differences between the two groups in the outer retinal layers (OPL, ONL, and PR) in either the central or peripheral retina. We did not find a difference in the PR layer, further confirmed by immunolabeling the photoreceptor outer segments with anti-rhodopsin in cross-sections (S1 Fig).

## L-HIPR shows abnormal retinal vascular development and persistent peripheral avascular retina

We next focused on the retinal vascular pathology and peripheral avascular area, using retinal flat-mounts labeled with Isolectin B4 (IB4) (Fig 2A). Compared to room air control eyes, we

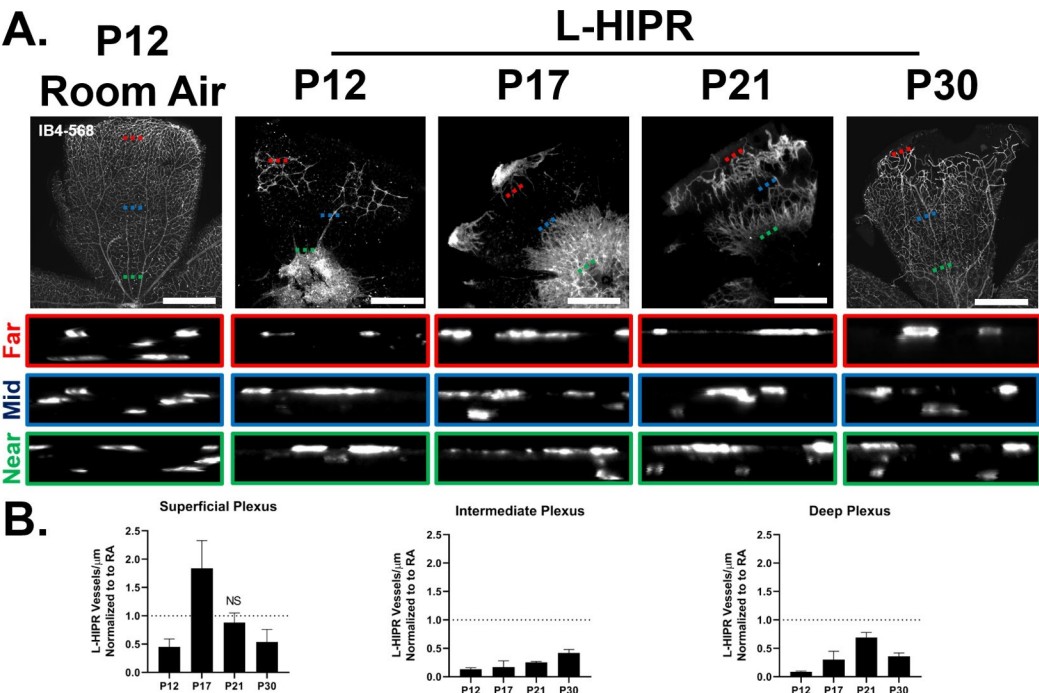

**Fig 2. L-HIPR causes delayed retinal vascular development.** (A) Retina flat-mount samples ($n = 3$) from both room air and L-HIPR groups at P12, P17, P21 and P30 were stained with aIB4 to detect the vascular development. There is sporadic vascularization at P12 in L-HIPR and by P30 the vasculature fails to completely reach the retinal margin. L-HIPR vessels also appear tortuous and dilated, particularly at P17 and P21. XZ spinning disc confocal line scans of IB4 labeled retinal flat-mounts were acquired to visualize the delayed vascular plexuses in L-HIPR. Scans were taken near the optic nerve, mid-retina, or periphery and relevant XZ optical scan insets are denoted by green, blue, and red dashed lines, respectively. (B) Quantification of blood vessel density within the 3 vascular plexuses in L-HIPR normalized to age matched room air control. Non-significant outcomes were marked as NS, all others are statistically significant, $p < 0.05$, two-sided $t$-test. Scale bars equal 300 μm.

found that retinal vascular development was markedly delayed in L-HIPR. At P12, few vascular structures were detected in the central retina in L-HIPR. Interestingly, at that stage we observed isolated pockets of IB4+ labeled blood vessels located in the mid-peripheral retina that did not appear to originate from the optic nerve. L-HIPR retinal vasculature spread radially from the optic nerve to the mid-peripheral retina by P17, with additional islands of disconnected peripheral vessels. By P21, the L-HIPR retina vascularization was nearly continuous, although two distinct vascular fronts could be observed (Fig 2). Retinal vascularization was ultimately continuous at P30 with residual avascular areas in the periphery.

## The intermediate plexus is developmentally delayed in L-HIPR

In a mature mouse retina, the blood vessels are localized to three vascular plexuses with the superficial layer located in the GCL, the intermediate located in the IPL and the deep at the outer border of the INL. Developmentally, the superficial plexus reaches the retinal periphery by P7, followed by the deep and lastly the intermediate, which is completed by P13. Remarkably, retinal thickness assay in L-HIPR (Fig 1C and S2 Table) showed that the layers of the retina most affected by L-HIPR (GCL, IPL, INL) were these associated with the inner retinal vasculature.

Given the vascular defects observed in Fig 2A, we decided to quantify the density of the vascular plexuses. Normalizing L-HIPR values to their room air age-matched counterparts

revealed a significantly less dense vasculature in every plexus at each time point in L-HIPR except the superficial plexus at P17 and P21. Of particular interest, the intermediate plexus in L-HIPR was the most dramatically impaired, falling below 50% of the density of its room air counterpart at every timepoint (Fig 2B and S3 Table).

## Abnormal vascular development and delayed perfusion in L-HIPR

To understand whether decreased retinal perfusion could explain the observed thinning of the relevant retinal layers in L-HIPR, we performed cardiac perfusion with Alexa-488 fluorescent conjugated IB4 (IB4-488) to highlight *in vivo* perfused vessels. Retinal flat-mounts were then examined in triplicates, counterstained with Alexa-568 conjugated IB4 (IB4-568) label to highlight all retinal blood vessels. At P17 in room air animals, perfusion was already continuous in all vessels to the retinal periphery (S2A Fig). Next, we focused on the L-HIPR retina. As shown in Fig 3A, we observed no detectable retinal perfusion at P17. At P21, we identified blood vessels that were co-labeled with IB4-488 and IB4-568 in the central retina whereas the peripheral vessels were not perfused. Interestingly, there were prominent perfused vessels running circumferentially in the retinal mid-periphery (Fig 3A P21 inset, asterisk). By P30, nearly all vessels are perfused in a continuous, albeit somewhat disorganized, fashion from the optic nerve to the retinal periphery.

To investigate whether these non-perfused vessels were undergoing vascular regression, we labeled the L-HIPR flat-mounts with an antibody against endothelial cell–specific molecule 1 (Esm1), a tip cell marker. In Fig 3B, we observed Esm1-positive tip cells on either side of, and

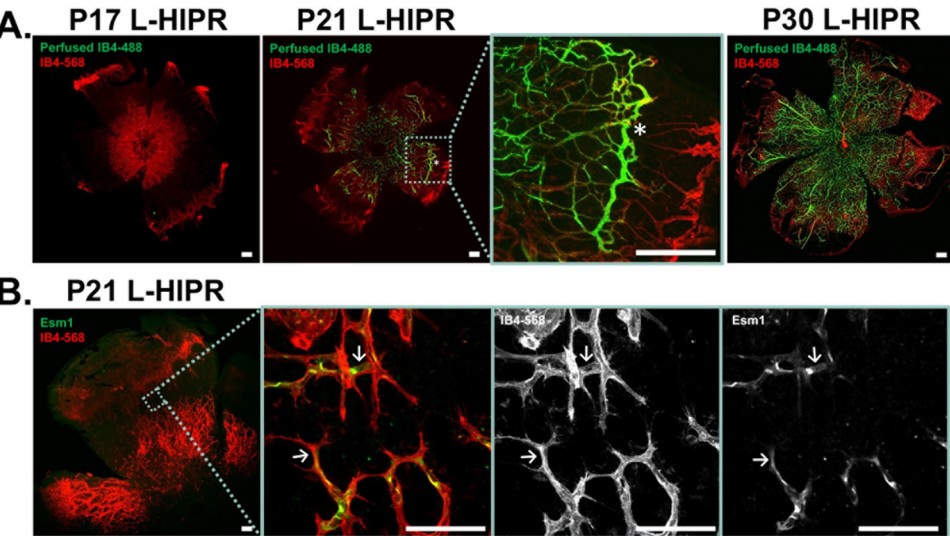

**Fig 3. Anomalous retinal vascular development, with two opposing angiogenic fronts, and transient circumferential vasculature is observed in L-HIPR.** (A) Alexa488 conjugated IB4 (green) was delivered in vivo via cardiac perfusion to highlight functional blood vessels. Retinal flat-mounts were prepared with Alexa568 conjugated IB4 (red) labeling all blood vessels. At P17 in L-HIPR no green signal is seen signifying that these vessels were not perfused. By P21 L-HIPR, perfusion is seen in the central retina, but peripheral vessels remain non-perfused. The light blue inset at P21 L-HIPR shows a region of interest displaying abnormal blood vessels traversing the retina circumferentially (marked with asterisk), which we suspect is persistent hyaloidal vasculature invading the retina. P30 L-HIPR shows mostly perfused retina from the optic nerve to periphery with resolution of the circumferential vasculature. (B) P21 L-HIPR flat-mounts immunolabeled with the tip-cell marker Esm1 (green) and blood vessels marked with IB4 (red). The light blue insert shows a region of interest where central and peripheral blood vessel fronts display IB4+ filopodia and Esm1+ tip-cells, indicating two angiogenic fronts that meet in the retinal mid-periphery. Scale bars equal 500 μm.

pointing towards, the avascular zone in the mid-peripheral retina. The vessels at the leading edge of the vascular fronts also displayed filopodia in the IB4-568 channel, which, together with the Esm1-positive cells confirm that the vessels are in a pro-angiogenic state. Furthermore, it would suggest that the vessels in the periphery are migrating inwards towards the tip cells and vessels arising from the optic nerve.

## The hyaloidal vessels persist and vascularize the retina in L-HIPR

To further examine the persistent hyaloidal vessels in L-HIPR, we evaluated retinal cross sections (Fig 4). The hyaloidal vessels were absent, with normal retinal vascular stratification visible by P12 in the room air eyes. In L-HIPR, the hyaloidal vessels persisted well into P21 and started to invaginate into the peripheral retina at P12, and the midperipheral retina at P17. These findings, coupled with the isolated peripheral pockets of blood vessels seen in the flat-mounts (Fig 2A) as well as the circumferential IB4-488 perfused vessels in the midperipheral retina, and the 2 fronts of angiogenesis (Fig 3B) suggest that the hyaloid vessels are participating in angiogenesis and contributing to the rescue of the largely avascular retina observed at P12 in L-HIPR.

## Inflammation in the L-HIPR retina

To expand upon the pathological consequences in this model, we examined markers of retinal inflammation in L-HIPR. In Fig 5A, frozen cryosections were labeled with the macrophage marker F4/80 and glial fibrillary acidic protein (GFAP), marking activated Muller cell glia. In the room air cryosections at P21 and P30, little to no F4/80 signal was observed and GFAP signal was limited to the astrocytes in the GCL. In contrast, in the L-HIPR tissue, F4/80+ macrophages were abundant at both P21 and P30. Furthermore, GFAP signal was seen in long vertical streaks, indicating activated Muller cells. Together, these findings indicate that retinal inflammation is prominent between P17 and P21 in the L-HIPR retina.

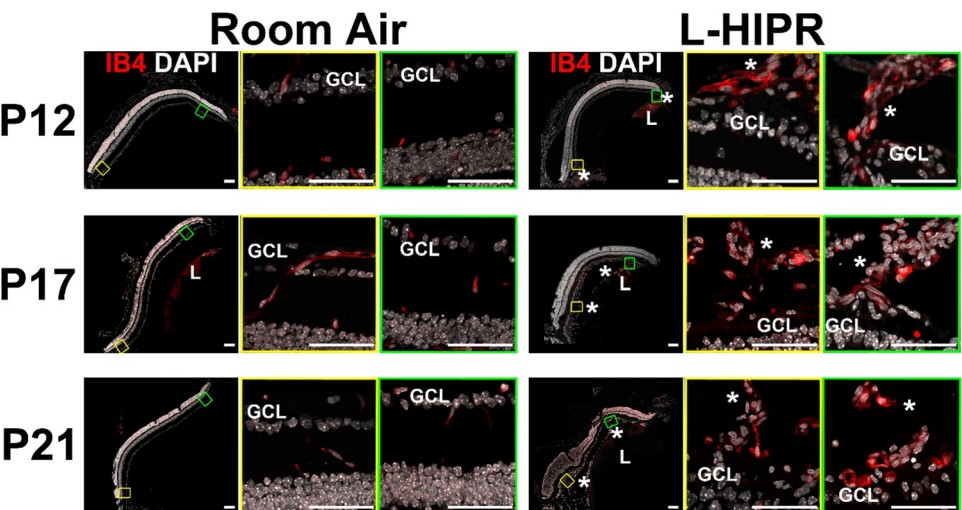

**Fig 4. The hyaloidal vessels persist and rescue the retinal vasculature in L-HIPR.** Retinal cross sections in room air and L-HIPR groups (*n* = 3) were stained with Isolectin B4 (IB4; red), and DAPI (white) at P12, P17 and P21. Central (yellow box) and peripheral (green box) zoomed retinal cross-sections are presented. The hyaloidal vessels were absent, with normal retinal vascular stratification by P12 in the room air eyes. In L-HIPR, the hyaloidal vessels persisted well into P21 and started to invade the peripheral retina at P12 (Asterisks). The later timepoints further illustrate how the hyaloidal vessels invade the mid-retina at P17 and P21. Scale bar 50 μm, GCL: ganglion cell, *: hyaloid, L: lens.

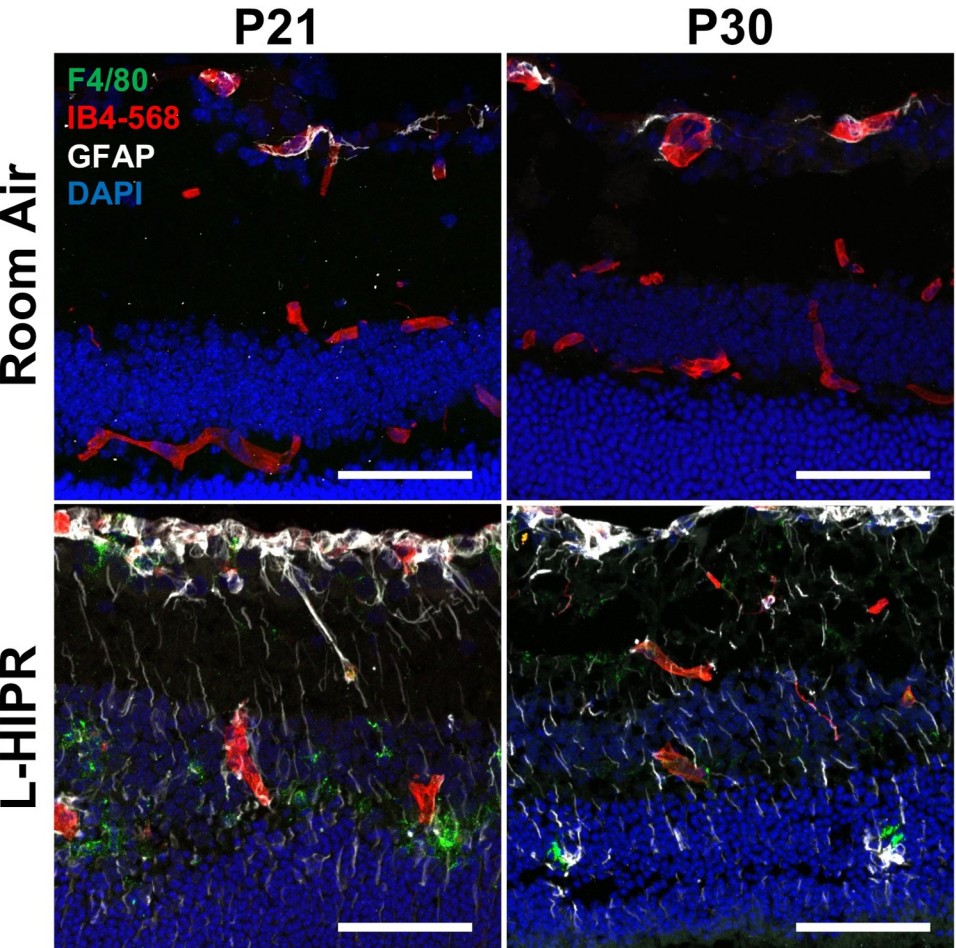

**Fig 5. The L-HIPR retina is inflamed.** (A) Immunolabeled cryosections at P21 and P30 of room air and L-HIPR tissue. Macrophages were labeled with anti-F4/80 (green), blood vessels with IB4 (red), astrocytes/activated Muller cells with GFAP (white), and nuclei with DAPI (blue). The room air tissues show GFAP⁺ astrocytes in the ganglion cell layer, however, the Muller cells become GFAP⁺ indicating an activated state in L-HIPR at both P21 and P30. F4/80⁺ macrophages can be seen in the inner and outer nuclear layers in the L-HIPR sample. Scale bars equal 50 µm.

### Decreased vascular density and pericyte counts in L-HIPR

We focused on P21 and P30 timepoints using flat-mounts stained with CD31 to label the endo-thelial cells and NG2 to label the pericytes (Fig 6A). Earlier timepoints were not considered due to the extensive vascular disorganization observed and the technical challenges in identify-ing and quantifying pericytes. When we analyzed the overall projected vascular density, and consistent with our findings in cross sections, the L-HIPR retinas displayed decreased vessel density at P21 and P30 (P21 $p<0.05$; P30 $p<0.001$) (Fig 6B). Whereas the P21 pericyte cover-age was not significantly different between L-HIPR and RA, at P30 L-HIPR flat-mounts showed 34% lower pericyte density compared to room air ($p<0.001$) (Fig 6C), suggesting either pericyte degeneration or defective development and attachment.

### L-HIPR is associated with pre-retinal fibrosis

We wanted to explore whether the retinal vascular maldevelopment in L-HIPR was associated with a fibrotic reaction. We used Masson trichrome stain to stain collagenous fibrotic tissue

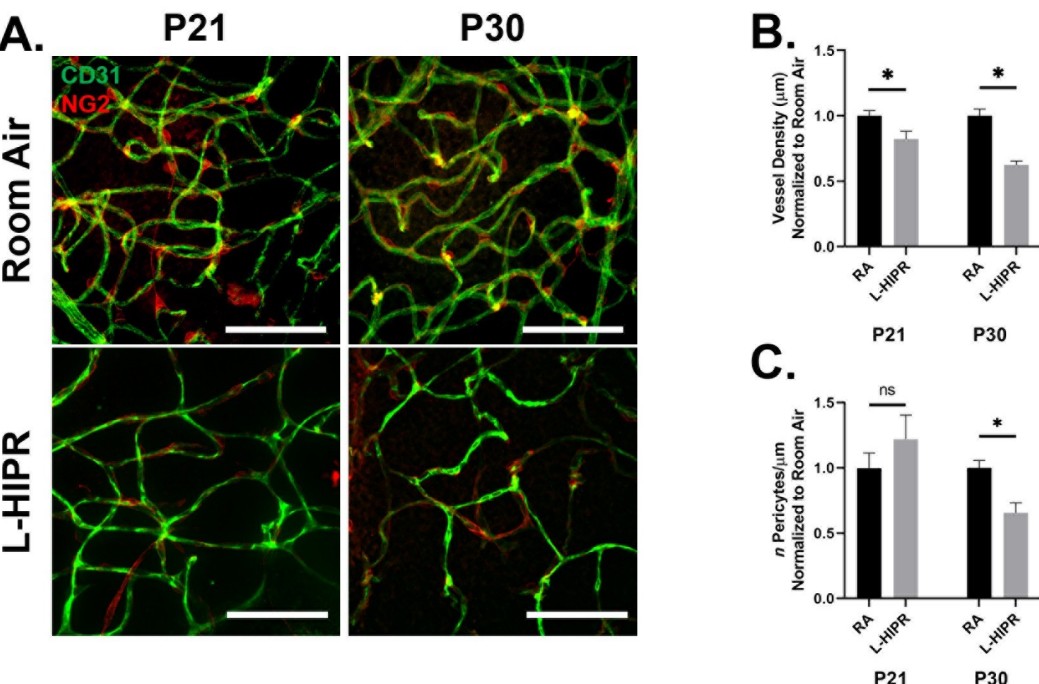

**Fig 6. Lower pericyte count is observed in L-HIPR.** (A) Immunolabeled retinal flat-mounts at P21 and P30 of room air and L-HIPR tissue. Blood vessels were labeled with anti-CD31 (green), and pericytes with NG2 (red). Scale bars equal 50 μm. (B) Graphical representation of vessel density where CD31 vessel length after skeletonization was quantified against image area and normalized to room air values. At both time points the L-HIPR vascular density was significantly less dense. (C) Graphical representation of NG2$^+$ pericyte counts per micron of CD31 vessel length (μm) normalized to room air values. There was not a statistically significant difference in pericyte counts between room air and L-HIPR at P21, however, there was a significant reduction at P30.

(Fig 7A). These images show collagenous tissue in the mid-vitreous, surrounding the hyaloidal vessels in P21 and P30 L-HIPR eyes. We also found that αSMA antibody highlighted islands of fibroblastic proliferation on the retinal surface at P21 and P30, which were distinct from the retinal vasculature. (Fig 7B). Compared to extensive αSMA expression in the L-HIPR, room air retinas expressed αSMA only in the retinal vasculature, consistent with smooth muscle cells mainly in the arterioles (S2 Fig). Both trichrome and αSMA show that the scar tissue formation is more extensive in P30 (Fig 7A and 7B). Retinal αSMA expression in the L-HIPR eyes shows that fibroblastic proliferation occurs on the retinal surface in addition to the fibrous bands in the vitreous stained by trichrome (Fig 7A and 7B).

## Discussion

This study expands our understanding of the L-HIPR model, originally reported by McMenamin et al.(6) We further quantified the timeline of the vitreoretinal pathologic changes to expand upon the previously reported delayed vascular development and persistence of hyaloidal vessels.(6) In addition, we characterized hyaloidal rescue of the retinal vasculature and the novel features of inflammation and pericyte defects in this model.

These observations place this model in the middle of a spectrum of rodent neonatal oxygen exposures designed to model neonatal prematurity and its sequelae, with oxygen induced retinopathy as the most well-characterized and least disruptive to retinal development.(2) On the severe end of the spectrum of these rodent models, our group previously reported the model of hyperoxia-induced proliferative retinopathy (HIPR), where neonatal mice are exposed to 75%

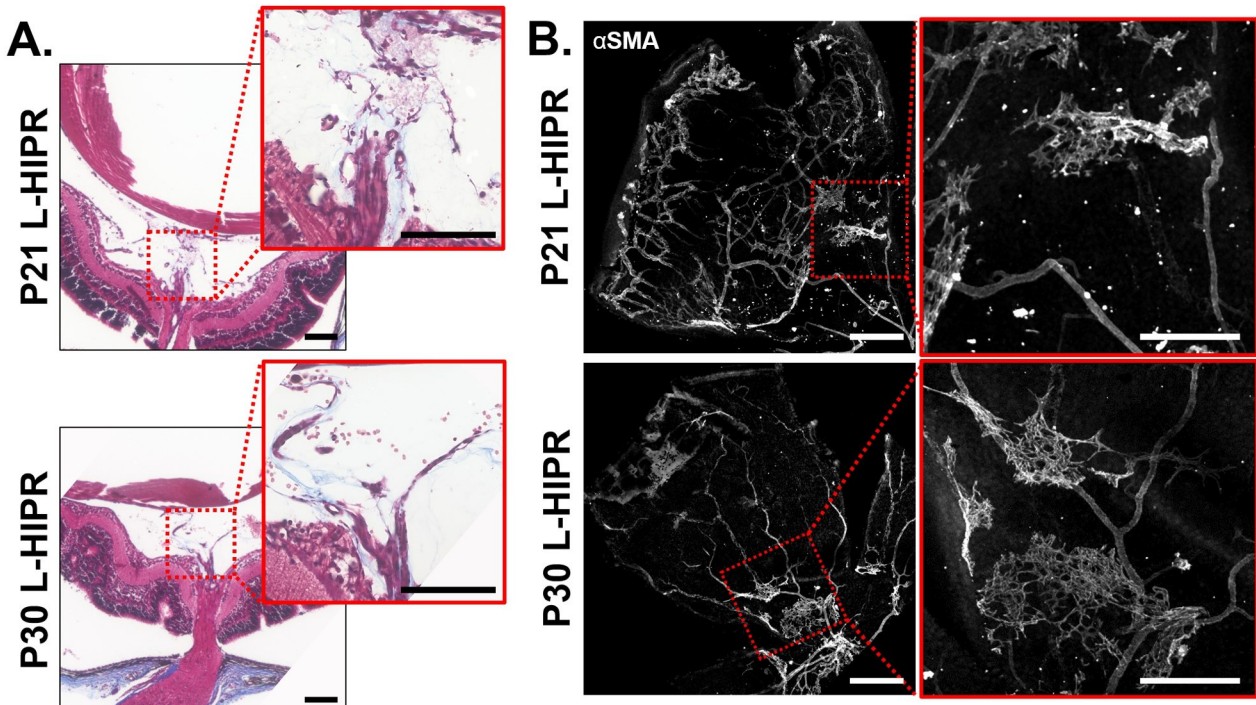

**Fig 7. L-HIPR model shows preretinal and retinal fibrosis and scarring.** (A) Masson trichrome staining of P21 and P30 L-HIPR tissue. Red inserts highlight regions of interest showing collagenous tissue (blue) in the vitreous surrounding the persistent hyaloidal vessels. (B) L-HIPR retinal flat-mount immunolabeled with α-SMA. Red inserts highlight regions of interest showing extravascular islands of α-SMA immunolabeling suggesting fibroblastic cellular proliferation on the retinal surface. Scale bars equal 300 μm.

oxygen from P0 to P14, an exposure that models neonatal bronchopulmonary dysplasia [9]. We found that HIPR demonstrated a severe phenotype of disrupted retinal vascular development, intra-retinal angiogenesis, oxidative stress, inflammation, and retinal detachment [9]. HIPR appeared to be an extreme form of retinal maldevelopment, unlike clinical ROP. Compared to the HIPR phenotype, L-HIPR is a modified exposure using lower oxygen concentration (65%) and shorter duration (P0-P7), with apparently limited disruption of the neurovascular development. This places this model closer to the human chronic ROP phenotype, with the additional advantage over OIR that these changes persist long-term and do not resolve spontaneously, offering researchers the opportunity to study disease modifying therapies.

Despite the significantly delayed retinal vascular development in L-HIPR, we found that the central and peripheral retinal structure remained relatively preserved during the early time points (up to P17). By P30, we found that the central retinal lamination was disrupted while the overall peripheral retinal thickness was significantly decreased. We found that the central GCL became significantly thicker in L-HIPR at P17. Considering that hyaloidal vessels invaginated into the inner retina at P17, we hypothesize that this could be related to endothelial barrier disruption in these hyaloidal vessels, intraretinal leakage, or disruption associated with preretinal fibrosis and traction (Fig 7A). The central and peripheral IPL in L-HIPR were thinner than the room air group at p17, which may be related to the delayed formation of middle capillary plexus, which we characterized (Fig 1C). Interestingly, at P30 the IPL thickness was similar the control group in the central retina, possibly because of the relative normalization of the middle capillary plexus by that point, allowing the IPL to further develop. The outer retinal layers in the center and peripheral retina were not significantly affected, suggesting that the

choroidal vascular development is likely intact, preserving the outer retina. Relative preservation of the outer retina is also a distinct feature in the OIR model [10].

Overall, the retinal vascular development was delayed, and the vascular density was significantly decreased (Fig 3). The hyaloidal vessels persisted in the vitreous and invaginated the peripheral retina starting at P12. At that timepoint, retinal vascular development was delayed in L-HIPR with only central vascular islands while the hyaloidal vessels were much more substantial in size compared to the room air group. At P17, along with the invasion of hyaloidal vessels into the mid-peripheral retina, the retinal vasculature spread from the optic nerve to the retinal periphery. The hyaloidal vessels provides blood perfusion and nutritional support to the retina until the retinal vasculature develop [11]. As normal retinal development was delayed in L-HIPR, the requirement for hyaloidal perfusion likely underlies the persistent hyaloidal vessels in the L-HIPR model. With the help of the invading hyaloidal vessels, the retinal vasculature approaches the retinal periphery by P30, however it was much sparser and less well organized than in the room air group. When we evaluated the vessel density and pericyte coverage, was saw that pericyte numbers were decreased as well as the vessel density (Fig 6B). The lower number of pericyte could potentially be related to several factors, including defective development or attachment as well as pericyte degeneration. This issue will be important to resolve in future studies.

While the scope of our analysis focused on pathological and histological changes in this model, we can speculate that endothelial and non-endothelial signaling pathways contributed to the reprogramming of the hyaloidal and retinal vasculature. The signaling mechanisms controlling regression of the hyaloid are not fully understood. Interestingly, in a transgenic model with conditional deletion of VHL in retinal cells during development, a similar phenotype of hyaloid persistence and invagination into the retina was shown [12]. In that model, the pathogenesis was related to ectopic deep retinal VEGF expression as a result of anomalous oxygen sensing. Neurons are intricately involved in the process of hyaloidal regression. In one transgenic mouse study, neuronal vegfr2 was shown to be upregulated at birth allowing them to titrate and sequester the secreted vegf proteins. Deletion of neuronal vegfa or neuronal vegfr2 modulated the vitreous vegf levels in this model, leading to acceleration or delay of hyaloidal regression, respectively [13]. Further, a neuronal, light dependent dopamine pathway via opsin 5 has also been shown to be critical for the timing of hyaloidal regression [14]. An intricate interplay between the neurons, endothelial cell apoptosis and phagocytosis [15] as well as blood flow induced vasoconstriction [16] have also been reported. Further studies will be important to clarify the pathogenesis of hyaloidal invagination in this model, as it could present a potential therapeutic avenue for retinal vascular repair.

Along with the persistence of hyaloidal vessels, we found evidence of inflammation and subsequent scar tissue formation during L-HIPR. Inflammation was evident in the P21 and P30 L-HIPR cryosections by Muller cell activation and intraretinal macrophage infiltration, confirmed by F8/40 marker (Fig 3). We confirmed fibrosis using trichrome and $\alpha$SMA stains marked scar tissue and retinal fibrosis in the P21 and P30 L-HIPR (Fig 7). This finding is unique to this model, compared to the OIR, and provides a novel tool to investigate the development of preretinal fibrosis in ischemic retinopathies.

In conclusion, our study shows that limited high oxygen exposure (65% $O_2$) from P0 to P7 in L-HIPR replicates many aspects of human ROP (Fig 8). The dilated, maldeveloped vasculature strongly mimic the features of this disease. In addition, the avascular areas are largely in the peripheral retina, consistent with human ROP disease [17])., [18] Finally, we did not find any evidence of vascular repair and revascularization unlike the OIR model, with prolonged course of retinopathy in this model, and development of preretinal fibrosis making it ideal to study therapeutic interventions addressing these conditions. The persistence and stepwise

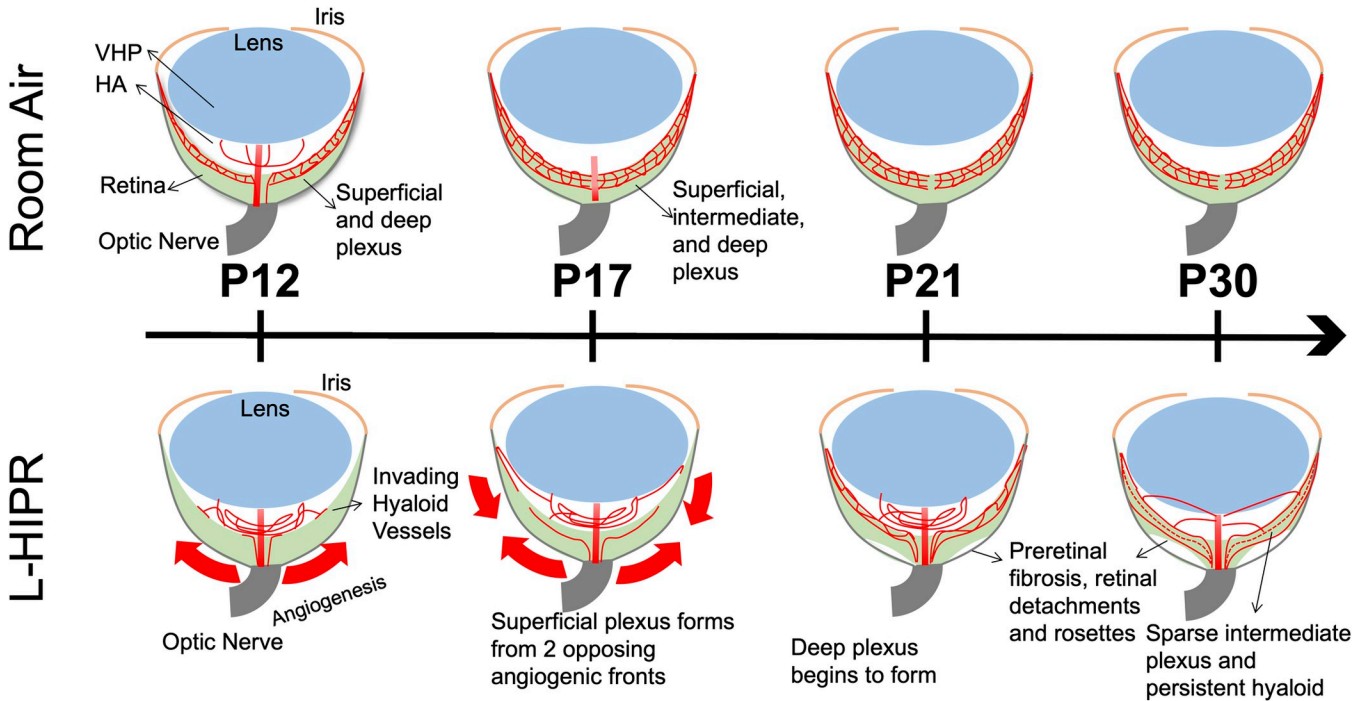

**Fig 8. Proposed model for the progression of L-HIPR.** Course of retinal vascular development, preretinal fibrosis and hyaloidal rescue of the retinal vasculature during L-HIPR exposure.

migration of hyaloidal vessels into the retina could be rescue for the inner retinal development. Studying the molecular mechanisms of hyaloidal-neuronal signaling in this model would be an important future goal to improve our understanding of L-HIPR, hyaloid regression and in turn ROP and other disorders such as persistent fetal vasculature.

## Supporting information

**S1 Checklist. The ARRIVE guidelines 2.0: Author checklist.**
(PDF)

**S1 Table. Effect of L-HIPR on weight gain.** Significantly lower weight gain was detected at P30 in L-HIPR group, but average litter weights were similar to room air controls before these neonates were weaned. Values presented as mean weight (g) ± S.E.M., *: $p < 0.05$, one-way ANOVA with Bonferroni post hoc analysis.
(TIF)

**S2 Table. Thickness quantification of all retinal layers.** We compared the room air and L-HIPR retinal layers at P12, P17, P21, and P30 by measuring central (near the optic nerve) and peripheral regions.
(TIF)

**S3 Table. L-HIPR blood vessel density values within each vascular plexus.** Quantification of blood vessel density within the 3 vascular plexuses in L-HIPR normalized to age matched room air control.
(TIF)

**S1 Fig. Normal outer segments of the photoreceptors seen in L-HIPR.** Immunolabeled paraffin cross-sections labeled with anti-Rhodopsin (green) and nuclei with DAPI (blue) overlayed with differential interference contrast microscopy. Scale bars equal 50 μm.
(TIF)

**S2 Fig. Representative normal images for room air eyes.** (A) IB4-488 cardiac perfusion (B) α-SMA and (C) Masson Trichrome staining. Scale bars equal 300 μm in (A) and (B), 50 μm in (C).
(TIF)

## Author Contributions

**Conceptualization:** Amani A. Fawzi.

**Formal analysis:** Thomas Tedeschi.

**Funding acquisition:** Amani A. Fawzi.

**Project administration:** Amani A. Fawzi.

**Resources:** Amani A. Fawzi.

**Supervision:** Amani A. Fawzi.

**Writing – original draft:** Wei Zhu.

**Writing – review & editing:** Kendal Lee, Amani A. Fawzi.

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
