## [Decision Letter · Decision Letter 0]

9 Mar 2022

PONE-D-22-02734Limited Hyperoxia Induced Proliferative Retinopathy: A Model of Persistent Retinal Vascular Dysfunction, Preretinal Fibrosis and Hyaloidal Vascular Reprogramming for Retinal RescuePLOS ONE

Dear Dr. Fawzi,

Thank you for submitting your manuscript to PLOS ONE. After careful consideration, we feel that it has merit but does not fully meet PLOS ONE’s publication criteria as it currently stands. Therefore, we invite you to submit a revised version of the manuscript that addresses the points raised during the review process.

The reviewers write that the study is of interest and has been thoroughly carried out. They ask for a number of textual clarifications, some missing data, and detailing of the underlying methodology, as outlined in the reviews below.

We look forward to receiving your revised manuscript.

Kind regards,

Roeland M.H. Merks, Ph.D

Academic Editor

PLOS ONE

Journal Requirements:

2. As part of your revision, please complete and submit a copy of the Full ARRIVE 2.0 Guidelines checklist, a document that aims to improve experimental reporting and reproducibility of animal studies for purposes of post-publication data analysis and reproducibility: https://arriveguidelines.org/sites/arrive/files/Author%20Checklist%20-%20Full.pdf (PDF). Please include your completed checklist as a Supporting Information file. Note that if your paper is accepted for publication, this checklist will be published as part of your article.

[This work was supported by a grant from the National Institutes of Health (1R01EY030121, A.A.F.), and the Illinois Society for the Prevention of Blindness (KL). Histology services were provided by the Northwestern University Mouse Histology and Phenotyping Laboratory which is supported by NCI P30-CA060553 awarded to the Robert H Lurie Comprehensive Cancer Center. Imaging work was performed at the Northwestern University Center for Advanced Microscopy generously supported by CCSG P30 CA060553 awarded to the Robert H Lurie Comprehensive Cancer Center.]

 [This work was supported by a grant from the National Institutes of Health (1R01EY030121, A.A.F.), and the Illinois Society for the Prevention of Blindness (KL). 

NO-The funders had no role in study design, data collection and analysis, decision to publish, or preparation of the manuscript.]

4. Please include your tables as part of your main manuscript and remove the individual files. Please note that supplementary tables (should remain/ be uploaded) as separate "supporting information" files.

Reviewers' comments:

Reviewer's Responses to Questions

**Comments to the Author**

1. Is the manuscript technically sound, and do the data support the conclusions?

Reviewer #1: Yes

Reviewer #2: Yes

2. Has the statistical analysis been performed appropriately and rigorously? 

Reviewer #1: Yes

Reviewer #2: Yes

3. Have the authors made all data underlying the findings in their manuscript fully available?

Reviewer #1: Yes

Reviewer #2: No

4. Is the manuscript presented in an intelligible fashion and written in standard English?

Reviewer #1: Yes

Reviewer #2: Yes

5. Review Comments to the Author

Reviewer #1: Very nice characterization of an OIR variant model, with similarity to clinical ROP.

no significant concerns with the science.

the introduction may benefit from more precise/specific wording, e.g. there are multiple OIR models but you are describing the original OIR model in mice - should be explicit. Also, dont overstate the differences between OIR and ROP - its true OIR usually regresses spontaneously, but so does ROP (> 90% of cases and even those with "threshold" ROP often dont detach).

Reviewer #2: In this study, Tedeschi et al. presented a novel mouse model of proliferative retinopathy, in which newborn mice with dams exposed to 65% O2 from p0 to p7. The authors performed an extensive morphological analysis on those animals and observed L-HIPR exposed mice developed avascular retina, preretinal fibrosis, persistent hyaloidal vessels, retinal inflammation and reduced pericyte recruitment. This is an interesting study; however, I have some concerns listed below.

1. Page6, Line 100 to 102: “Dams were rotated between hyperoxia and room air pups”

What was the survival rate of pups if dams were switched?

2. Page 11, Line 198-200: “ Throughout these time points, the L-HIPR animals did not display any differences in developmental progression or behavior”

Can the authors provide the details of the approaches used to measure developmental progression or behavior?

3. Page 21, Line 419-421: “at P30 L-HIPR flat-mounts showed 34% lower pericyte density compared to room air (p<0.001)”.

Can the authors provide the methods used to quantify pericyte density? Why did the authors believe less pericyte density was due to pericyte loss? Was it possible because of the defects in pericyte development or recruitment?

4. Page 24, Line 476- 478: “Compared to the HIPR phenotype, L-HIPR is a modified exposure using lower oxygen concentration (65%) and shorter duration (P0-P7), with apparently limited disruption of the neurovascular development.”

Have the authors performed ERG analysis on L-HIPR exposed animals? Compared to room air raised animals, did they show abnormal function of the retina?

5. There was no Supplemental Table 2

6. Overall, the quality of figures was very poor. Can the authors provide figures with better resolution? The text font size was not consistent, please correct it.

6. PLOS authors have the option to publish the peer review history of their article (what does this mean?). If published, this will include your full peer review and any attached files.

Reviewer #1: No

Reviewer #2: **Yes: **HAIBO WANG

---

## [Author Response · Author response to Decision Letter 0]

21 Mar 2022

Reviewer #1: Very nice characterization of an OIR variant model, with similarity to clinical ROP.

no significant concerns with the science.

the introduction may benefit from more precise/specific wording, e.g. there are multiple OIR models but you are describing the original OIR model in mice - should be explicit. Also, dont overstate the differences between OIR and ROP - its true OIR usually regresses spontaneously, but so does ROP (> 90% of cases and even those with "threshold" ROP often dont detach).

Author Response: Thank you for these insightful comments. We have made the requested changes regarding the OIR model as well as updating the differences between OIR and ROP. These edits are in the first paragraph of the revised manuscript Introduction.

Reviewer #2: In this study, Tedeschi et al. presented a novel mouse model of proliferative retinopathy, in which newborn mice with dams exposed to 65% O2 from p0 to p7. The authors performed an extensive morphological analysis on those animals and observed L-HIPR exposed mice developed avascular retina, preretinal fibrosis, persistent hyaloidal vessels, retinal inflammation and reduced pericyte recruitment. This is an interesting study; however, I have some concerns listed below.

1. Page6, Line 100 to 102: “Dams were rotated between hyperoxia and room air pups”

What was the survival rate of pups if dams were switched?

Author response: We generally observed a survival rate of greater than 70% of pups when dams were rotated between hyperoxia and room air foster litters. This rotation also allowed for 100% survival rate of the dams. Unfortunately, without rotation, prolonged hyperoxia results in significant maternal weight loss and/or lethality during the experiment, creating significant additional impediments for pup survival. 

We have added this clarification to the manuscript Methods section, under the "Limited hyperoxia-induced Retinopathy L-HIPR" sub-heading.

2. Page 11, Line 198-200: “ Throughout these time points, the L-HIPR animals did not display any differences in developmental progression or behavior”

Can the authors provide the details of the approaches used to measure developmental progression or behavior?

Author response: We monitored developmental progression by weighing pups at each experimental timepoint in both L-HIPR and room air pups. Additionally, we did not notice differences between room air and L-HIPR pups in reaching observable developmental timepoints such as ear flaps starting to come away from the head (~postnatal day 3 [P3]), milk spot disappearance (~P6), progression of fur coverage (~P7-10), and eyelid opening (~P13). We also noticed normal developmental behaviors of L-HIPR litters such as surface righting (~P4-5), starting to eat solid food on the floor of the cage (~P14), and venturing away from the nest (~P14).

This clarification is now included in the Results section, under the first sub-heading “Long-term weight gain was suppressed in L-HIPR”

.

3. Page 21, Line 419-421: “at P30 L-HIPR flat-mounts showed 34% lower pericyte density compared to room air (p<0.001)”.

Can the authors provide the methods used to quantify pericyte density? Why did the authors believe less pericyte density was due to pericyte loss? Was it possible because of the defects in pericyte development or recruitment?

Author response: To quantify pericyte density, we had two independent, masked investigators count immunofluorescent labeled NG2+ marked pericyte cells in all three vascular beds in 60 individual regions of interest in 12 mouse retinas. We then standardized the pericyte count to the length of blood vessels in the same field of view. The length of blood vessels was quantified by skeletonizing immunofluorescent labeled CD31+ marked endothelial cells, as surrogate for vessel length. 

This has been further clarified in the Methods section, under “Quantifying vascular density and pericyte counts” sub-heading. 

We also agree with the reviewer that the reduction of pericyte density in L-HIPR could be due to defects in pericyte development, recruitment, or pericyte cell death. We ultimately hope to elucidate the mechanisms of L-HIPR induced pericyte loss in future works. 

We have now acknowledged these alternative possibilities in the Results section under subheading “Decreased vascular density and pericyte counts in L-HIPR” as well as in the Discussion , paragraphs 1 and 4. 

4. Page 24, Line 476- 478: “Compared to the HIPR phenotype, L-HIPR is a modified exposure using lower oxygen concentration (65%) and shorter duration (P0-P7), with apparently limited disruption of the neurovascular development.”

Have the authors performed ERG analysis on L-HIPR exposed animals? Compared to room air raised animals, did they show abnormal function of the retina?

Author response: We did not perform ERG analysis at any of our timepoints. ERG abnormalities have already been documented in the original study by McMenamin et al. 2014. These authors reported abnormal long-term neuroretinal functional changes in L-HIPR mice when compared to room air mice at 8 weeks of age. We therefore chose not to repeat these studies and expand our manuscript beyond what has already been reported.

5. There was no Supplemental Table 2

Author response: We apologize for omitting Supplemental Table 2 which shows thickness quantifications of all retinal layers at each experimental timepoint.

 The revised submission includes this under Supporting information, S2 Table.

6. Overall, the quality of figures was very poor. Can the authors provide figures with better resolution? The text font size was not consistent, please correct it

Author response: We apologize for this, as our original upload contained corrupted files. Please see the revised version. The reviewer may need to click on the link at the top right-hand corner of the pdf to download the full-sized, high-quality versions of these figures. 

We have also addressed the font issues in the revision. Thank you for your careful and insightful review.

---

## [Decision Letter · Decision Letter 1]

12 Apr 2022

Limited hyperoxia-induced proliferative retinopathy: A model of persistent retinal vascular dysfunction, preretinal fibrosis and hyaloidal vascular reprogramming for retinal rescue

PONE-D-22-02734R1

Dear Dr. Fawzi,

We’re pleased to inform you that your manuscript has been judged scientifically suitable for publication and will be formally accepted for publication once it meets all outstanding technical requirements.

Kind regards,

Roeland M.H. Merks, Ph.D

Academic Editor

PLOS ONE

Additional Editor Comments (optional):

Reviewers' comments:

Reviewer's Responses to Questions

**Comments to the Author**

1. If the authors have adequately addressed your comments raised in a previous round of review and you feel that this manuscript is now acceptable for publication, you may indicate that here to bypass the “Comments to the Author” section, enter your conflict of interest statement in the “Confidential to Editor” section, and submit your "Accept" recommendation.

Reviewer #2: All comments have been addressed

2. Is the manuscript technically sound, and do the data support the conclusions?

Reviewer #2: Yes

3. Has the statistical analysis been performed appropriately and rigorously? 

Reviewer #2: Yes

4. Have the authors made all data underlying the findings in their manuscript fully available?

Reviewer #2: Yes

5. Is the manuscript presented in an intelligible fashion and written in standard English?

Reviewer #2: No

6. Review Comments to the Author

Reviewer #2: The authors have addressed my comment properly. the reviewer does not have any concerns about research ethics or publication ethics.

7. PLOS authors have the option to publish the peer review history of their article (what does this mean?). If published, this will include your full peer review and any attached files.

Reviewer #2: No

---

## [Editor Report · Acceptance letter]

18 Apr 2022

PONE-D-22-02734R1 

Limited hyperoxia-induced proliferative retinopathy: A model of persistent retinal vascular dysfunction, preretinal fibrosis and hyaloidal vascular reprogramming for retinal rescue 

Dear Dr. Fawzi:

I'm pleased to inform you that your manuscript has been deemed suitable for publication in PLOS ONE. Congratulations! Your manuscript is now with our production department. 

Kind regards, 

on behalf of

Prof.dr. Roeland M.H. Merks 

Academic Editor

PLOS ONE